# In-utero exposure to PM$_{2.5}$ and adverse birth outcomes in India: Geostatistical modelling using remote sensing and demographic health survey data 2019–21

Arup Jana[1], Malay Pramanik[2], Arabinda Maiti[3], Aparajita Chattopadhyay[4], Mary Abed Al Ahad[5]*

1 Department of Humanities and Social Sciences, Indian Institute of Technology Delhi, Hauz Khas, New Delhi, India, 2 Urban Innovation and Sustainability, School of Environment, Resources and Development (SERD), Asian Institute of Technology (AIT), Klong Luang, Pathumthani, Thailand, 3 Postdoctoral Fellow, School of Geography, Newman Building, University College Dublin, Belfield, Dublin, Ireland, 4 Department of population & Development, International Institute for Population Sciences, Mumbai, India, 5 Associate Lecturer, School of Geography and Sustainable Development, University of St Andrews, St Andrews, Scotland, United Kingdom

* maaa1@st-andrews.ac.uk

## Abstract

This study investigates the influence of air quality on birth weight and preterm birth. Utilizing data from the national family health survey and raster images, the study employs various statistical analyses and spatial models to elucidate the connection between in-utero exposure to air pollution and birth outcomes, both at the individual and district levels. It was observed that approximately 13% of children were born prematurely, and 17% were born with low birth weight. Increased ambient particulate matter 2.5 concentrations during pregnancy were associated with higher odds of low birth weight (AOR: 1.4; 95% CI: 1.29–1.45). Mothers exposed to particulate matter 2.5 during pregnancy had a heightened likelihood of delivering prematurely (AOR: 1.7; 95% CI: 1.57–1.77) in comparison to unexposed mothers. Climatic factors such as rainfall and temperature had a greater association with adverse birth outcomes. Children residing in the Northern districts of India appeared to be more susceptible to the adverse effects of ambient air pollution. Employing a distributed spline approach, the study identified a discernible upward trend in the risk of adverse birth outcomes as the level of exposure increased, particularly following an exposure level of 40 particulate matter 2.5 ug/m$^3$. A 10 µg m$-3$ increase in particulate matter 2.5 exposure was associated with a 5% increase in the prevalence of low birth weight and a 12% increase in preterm birth. Among the different spatial models used in this study, the multiscale geographically weighted regression spatial model showed the best fit to the actual scenario, effectively capturing the spatial relationships between particulate matter 2.5 exposure and adverse birth outcomes. In addition to addressing immediate determinants such as nutrition and maternal healthcare, it is imperative to

**Data availability statement:** The study uses secondary data that are available on reasonable request through https://dhsprogram.com/data/dataset_admin/.

**Funding:** The author(s) received no specific funding for this work.

**Competing interests:** The authors have declared that no competing interests exist.

collaboratively address distal factors encompassing both indoor and outdoor pollution to attain lasting enhancements in child health.

## 1. Introduction

Ambient air pollution poses an existential global environmental threat to planetary and human health, with a disproportionate burden of its detrimental effects falling on those residing in low and middle-income countries [1]. Consequently, the United Nations Climate Change Conference has urged developed countries to provide financial support to less developed and developing countries to confront the adverse impacts of air pollution and establish appropriate mechanisms to mitigate climate change [2]. Besides being a significant driver of climate change, air pollution is the most critical risk factor for adverse health consequences [3]. Referred to as the "silent killer," ambient air pollution is among the top five risk factors for mortality in both males and females [4,5]. In 2019 alone, ambient particulate matter pollution was responsible for 118 million disability-adjusted life years (DALY) and 4.14 million deaths in 2019 [6]. Among air pollutants, ambient fine particulate matter 2.5 ($PM_{2.5}$) is considered the most harmful air pollutant [7,8]. These particles primarily originate from the burning of fossil fuels and biomass [9]. With a diameter of less than 2.5 microns, exposure to these particles increases the risk of respiratory diseases, lung cancer, stroke, and heart disease [8,10].

In the 2023 World Air Quality Report, India was ranked as the third most polluted country out of 134 nations based on its average yearly $PM_{2.5}$ levels [11]. Notably, about 7 out of 10 Indians are exposed to air pollution levels that exceed the national standard of 40 μg m−3 [12]. Study reported that about a million deaths in India were attributed to ambient particulate matter pollution in 2019 [13]. The key factors of this alarming trend are primarily attributed to the rising air pollution levels in the country due to urbanization and industrialization [1]. To address this critical air pollution situation, the government of India introduced the National Clean Air Program in 2019, setting a targeted reduction in air pollution. It envisages reaching a minimum 20% reduction in particulate matter concentration by 2024, compared to 2014 levels [14,15]. However, despite these significant efforts, India still ranks as the most polluted country in the world [14].

While air pollution has adverse impacts on the health of individuals across all age groups, infants and children are considered more vulnerable due to their developing organs and higher air intake per unit of body weight [16,17]. Ambient air pollution has been associated with a range of pediatric morbidities, including adverse birth outcomes (ABOs), asthma, cancer, and an increased risk of chronic diseases in life stages [8,18,19]. Among the ABOs associated with ambient air pollution are preterm birth (PTB), low birth weight (LBW), Stillbirth, gestational age, abortion and birth defects [20,21]. PTB and LBW are important predictors of under-five mortality and malnutrition [22,23]. The World Health Organization estimates that approximately 15 million babies are born preterm (<37 weeks) each year [23]. Furthermore, 15% of

babies worldwide are born with LBW, defined as <2500 grams at birth [24]. While LBW is a global public health concern, its prevalence is particularly higher in low and middle-income countries [25]. A study reported that a 10 μg m−3 increase in $PM_{2.5}$ reduced birth weight significantly in the first (−2.6 g) and third trimesters (−3.1 g) [26]. During the first trimester, $PM_{2.5}$ exposure above 21.36 μg m−3 was significantly associated with higher odds of adverse outcomes (AOR: 1.29, 95% CI: 1.00–1.67) [27]. However, it has been reported by the previous studies that factors like access to prenatal care, improved nutrition, and air quality interventions, such as using air purifiers and limiting outdoor activity, can help mitigate these adverse effects [28]. A systematic review of 41 studies found that exposure to $PM_{2.5}$ is associated with PTB, LBW, and small-for-gestational-age births [29]. A prospective cohort study conducted in Tamil Nadu provided the first quantitative evidence linking rural-urban $PM_{2.5}$ exposures during pregnancy of LBW [30]. This study concluded that increased exposure to particulate matter exposure during pregnancy is associated with a decrease in birth weight among newborns [30]. However, research in developing countries, albeit limited, established that maternal exposure to ambient air pollution is associated with higher odds of LBW and PTB among children [31].

The mechanisms behind PTB and LBW due to exposure to $PM_{2.5}$ are not clearly understood. Some epidemiological and toxicological studies proposed different pathways to explain the paradox. Due to the finer size of $PM_{2.5,}$ inhaling particulate matter deposits in the lungs and affects the circulatory system [32]. Which is a reason of having oxidative stress, blood coagulation and placental inflammation that restricts the fetal growth [33,34]. Nevertheless, the presence of particulate matter in the human body disturbs oxygen transport and causes hormone dysfunction, which is the reason for placental insufficiency [35,36].

The majority of studies investigating the association between ambient air pollution with ABOs have primarily been conducted in high-income countries [31]. The evidence regarding this particular issue is somewhat less in developing countries [28]. According to National Family Health Survey-5, 18% of children born in the five years preceding the survey had low birth weight. India has been identified as a significant contributor to global preterm births [37]. Moreover, ambient air pollution levels are expected to increase in India as the country is transitioning from a rural economy to an urban one, with the urban population projected to reach 53% by 2050 from 35.87 in 2022 [38]. Despite the alarming rise in air pollution levels in India, there has been a paucity of research exploring its impact on ABOs. Moreover, the findings of a spatial analysis on the effects of air pollution on ABOs will be helpful for implementing area-specific schemes for policymakers or stakeholders, which has not yet been done at the national level in India. Therefore, the study hypothesizes that there is a significant spatial variation in the impact of ambient air pollution on LBW and PTB across different regions of India, with specific areas showing higher vulnerability to these ABOs due to elevated pollution levels. In order to address this research gap in the available evidence, the present study endeavors to investigate the impact of ambient air pollution on ABOs, particularly focusing on LBW and PTB at the national level. Further, the study utilized different geospatial models to highlight the vulnerable areas that need to be focused on, with robust estimated outcomes controlling for environmental, demographic, and socioeconomic confounding factors. The study utilizes nationally representative data from the latest round of the NFHS, 2019–21, thus ensuring the generalizability of results [39]. The results of this study might be imperative as LBW and PTB continue to be significant public health concerns in India. Moreover, this study will augment the existing but limited literature focused on investigating the association of ambient air pollution with LBW and PTBs, a question of paramount public health significance.

## 2. Data and methods

### 2.1. Ethics statement

The Department of Humanities and Social Sciences, Indian Institute of Technology Delhi provided ethical approval to conduct this study. This study utilises secondary data from the Indian Demographic and Health Survey (DHS), also known as National Family Health Survey (NFHS) in India. We used published large scale national data where every respondent was anonymized in the data set itself. As it is not based on a primary survey- cases, we need not to do any anonymization

in the study as the data is already made in that fashion following all ethical protocols. Thereby, it is certified that all applicable institutional and governmental regulations concerning the ethical use of human volunteers were followed during the course of the survey. Given that this study is based on secondary data analysis, no consent form was required.

## 2.2. Population and health data

Population data were extracted from the fifth NFHS, conducted across 36 states and union territories at the national level. A stratified two-stage sampling procedure was employed in the survey design. Primary Sampling Units were villages in rural areas and census enumeration blocks in urban areas. Survey questionnaires were prepared to take out detailed information on maternal and child health, as well as socioeconomic data. Additionally, the NFHS data incorporated the collection of biomarker data [39]. The survey encompassed reproductive history, including data on contraceptive use, pregnancy, birth, termination, and abortion. Children born 0–5 years preceding the survey were considered in the research. The protocol for the NFHS-5 survey, including the content of all the survey questionnaires, received approval from both the International Institute of Population Sciences institutional review board and the ICF institutional review board. Furthermore, it underwent review protocol by the U.S. Centres for Disease Control and Prevention. Further details on the sampling design can be obtained in the Indian National Report (https://dhsprogram.com/pubs/).

## 2.3. Air pollution data

In this study, global level $PM_{2.5}$ data was derived from the Atmospheric Composition Analysis Group, to measure the exposure of mothers during the pregnancy period [40]. The spatial resolution of $PM_{2.5}$ was 0.01° × 0.01° and Resolution-Tiered Approach was employed to estimate the concertation of $PM_{2.5.}$ The data is prepared by the fusion of multiple sensors including Aerosol Optical Depth measurements from NASA's Moderate–Resolution Imaging Spectro-radiometers, Multi-Resolution Imaging Spectro-Radiometers, and Sea–viewing Wide Field–of–View Sensor instruments with the Goddard Earth Observing System and chemical transport model data. The data was further calibrated to match global ground-based observations using a Geographically Weighted Regression technique, and its accuracy has been validated in several previous studies [41,42]. Each survey participant was assigned an average in-utero air pollution exposure based on the child's date of birth and the duration of pregnancy, as recorded in the NFHS survey data. For instance, if a child was born in January 2019 with a pregnancy duration of 9 months, the in-utero exposure period was considered from May 2018 to the end of January 2019. The cluster points were randomly displaced by 2 and 5 km in urban and rural areas, respectively, to maintain the privacy of participants. Thus, a 3 km buffer was created around each cluster point for the extraction of pollutant data [43].

## 2.4. Climate data

We used the Climate Hazards Group InfraRed Precipitation with Station precipitation dataset, covering a span of 43 years from 1981 to the present [44], with a spatial resolution of 5 km (https://www.chc.ucsb.edu/data/chirps). This high-quality dataset served as the foundation for developing a comprehensive and robust climatic index, laying the groundwork for deeper insights into the long-term trends and patterns of climatic hazards in various geographical areas.

We also utilized the ERA5-Land product, distributed by the Copernicus Climate Change Service Climate Data Store of ECMWF, to analyze air temperature in our study [45]. This dataset provides monthly temperature values at a spatial resolution of 0.1 degrees (approximately 11,200 meters at the equator) on a latitude/longitude Climate Modelling Grid, covering the period from January 1981 onwards. The ERA5-Land product, along with its fine resolution and long-term availability, enables detailed assessments of air temperature variations and their implications for various ecosystems and human activities [44,46]. Exposure to precipitation and air temperature was calculated using the same methodology as $PM_{2.5}$ exposure.

## 2.5. Variable description

**2.5.1. Dependent variables.** According to the World Health Organization (WHO), preterm birth (PTB) is defined as a live birth occurring before completing 37 weeks of gestation, while low birth weight (LBW) refers to a birth weight of less than 2,500 grams [47]. In the present study, we followed the methodology outlined by Jana, Banerjee, and Khan (2023) [48], to measure PTB using the calendar method of the Demographic and Health Surveys [49]. The NFHS gathered data on birth weight through the following questions: "Was (child's name) weighed at birth?" and "How much did (child's name) weigh?". This information was recorded in two ways—either based on the mother's recall or documented using a card that recorded the baby's weight [39]. The other outcome variable in this study was preterm birth, which is estimated based on the duration of pregnancy. Both outcome variables are dichotomous, with '0' signifying that the child did not have LBW/ was not preterm, and '1' indicating that the child had LBW/was PTB.

**2.5.2. Model covariates.** We selected the factors associated with ABOs based on published literature and the availability of variables within the dataset. The study included the child's characteristics, such as the sex of the child (male and female) and birth order (1st, 2nd and 3rd & above), which were considered as confounding factors. Babies born at home was defined as non-institutional delivery and delivery taken place at any medical institution was considered as institutional delivery. The overall health of the mother is crucial in shaping the wellbeing of newborns [50]. NFHS-5 obtained anthropometric measurements through biomarker assessments. These measurements were used to calculate Body Mass Index (BMI) by dividing weight in kilograms by height in meters squared ($kg/m^2$). BMI was categorized into four groups; thin (BMI < 18.5), normal (BMI 18.5–24.9), overweight (BMI 25–30), and obese (BMI ≥ 30.0). The model incorporated several maternal variables, such as mother's level of education (categorized as illiterate/primary, secondary, and higher education) and mother's age at the time of childbirth (< 20, 20 – 24, 25–29, and 30 years & above). The survey asked respondents about their primary cooking fuel source. Wood, coal, dung cake, and crop residue were categorized as solid or unclean fuels, while natural gas, liquefied petroleum gas, and electricity were classified as clean fuels. The household's wealth quintile was categorized into three categories; 'poor', 'middle' and 'rich'. Additionally, the household's religion was recoded as 'Hindu', 'Muslim' and 'Others' (such as Sikh, Christian, Jain etc.).

## 2.6. Statistical analysis

The weighted prevalence of LBW and PTB was estimated using the exposed sample, and the Chi-square test (χ2) was performed to evaluate the association between dependent and independent variables. Geospatial analysis has been conducted using ArcGIS (version 10.8) to prepare the prevalence maps. Further, multivariate logistic regression was employed to examine the association between exposure to $PM_{2.5}$ during pregnancy and birth outcomes after controlling all the possible confounding factors. The logistic regression model is defined as,

$$logit(p) = \log\left(\frac{p}{1-p}\right) = \beta_0 + \beta_1 * x_1 \ldots\ldots + \beta_k * x_k + \varepsilon$$

Where, $\beta_0$ is intercept and β1… βk are regression coefficients indicating the relative effect of a particular explanatory variable on the outcome, while $\varepsilon$ is an error term. Further, bivariate Local Indicators of Spatial Association (LISA) was used in the study to explore the spatial association between exposure to $PM_{2.5}$ and ABOs. A sensitivity analysis was employed by adjusting different determinants to explore the relation between in-utero exposure to $PM_{2.5}$ and birth outcomes such as PTB and LBW. Further, in the study, a cubic spline function was utilized to visually depict the connection. This connection involved the likelihood of occurrences such as having children with LBW and PTB, all influenced by exposure to air pollution during pregnancy. A spline function integrates several polynomial sections linked by knots, forming a continuous curve. The quantity of knots can be modified to control the curve and smoothness, increasing or decreasing it as needed. To determine knot placement, we employed quantiles of the exposure level of $PM_{2.5}$ in the individuals.

Bivariate LISA measures the local correlation between a variable and the weighted average of another variable in the neighborhood.

$$I_i = n_i \sum_j w_{ij} z_j$$

Where, $Z_i$ denotes standardized variable of interest and $W_{ij}$ is weight matrix. The bivariate LISA functionality is a powerful tool for exploring spatial relationships between two variables. It generates cluster maps that reveal the spatial association by identifying areas where the relationship between the two variables is statistically significant. A bivariate LISA cluster map is a specialized choropleth map that highlights spatial units with significant local Moran statistics, classifying them based on the type of spatial correlation observed. Specifically, bright red indicates high-high clusters, where high values of both variables are concentrated, and bright blue represents low-low clusters, where low values of both variables are grouped. Conversely, light red denotes high-low associations, where high values of one variable correspond to low values of the other, while light blue represents low-high associations, where low values of one variable are adjacent to high values of the other. High-high and low-low clusters suggest spatial clustering of similar values, while high-low and low-high clusters indicate spatial outliers, where neighboring values differ significantly. Additionally, the scatter plot of the dependent and independent variables for each spatial unit provides a statistical visualization of the association, complementing the cluster map by illustrating the nature and strength of the relationship between the two variables across the study area. Together, these tools enable a nuanced understanding of spatial patterns and their underlying associations, which is particularly valuable in geographic and spatial epidemiology studies.

Several geostatistical models were employed in this study to identify the best-fitting model for analyzing spatial patterns and associations. The results derived from this analysis provide critical insights that can guide policymakers in designing and implementing area-specific schemes, particularly targeting vulnerable regions identified through the models. These findings underscore the importance of evidence-based strategies for addressing regional disparities and improving resource allocation. First, Ordinary Least Squares (OLS) technique was used. This is a classic linear regression model that seeks to find the best-fitting line through the data by minimizing the sum of squared differences between observed and predicted values. OLS assumes that the relationship between the dependent and independent variables is constant across space. Second, we employed Geographically Weighted Regression (GWR) which is a spatial regression technique that allows for the exploration of spatially varying relationships between variables. It recognizes that the relationships may change across different geographic locations, providing a localized understanding of the data. Finally, Multiscale Geographically Weighted Regression (MGWR) which extends GWR by considering relationships at multiple spatial scales was applied [51]. This model accounts for varying relationships not only at the local level but also extends its analysis across different geographic scales, providing a more comprehensive analysis of spatial variability. The model employs two spatial regression models, namely the Spatial Lag Model (SLM) and the Spatial Error Model (SEM), each addressing distinct aspects of spatial dependency. The SLM is a type of spatial autoregressive model that considers the spatial dependency of observations. It assumes that the values of the dependent variable are influenced by the values of neighboring observations, incorporating spatial relationships into the model [52]. It accounts for the spatial interdependence of data points. On the other hand, the SEM is another spatial regression model that accounts for spatial autocorrelation in the error terms of the model. It assumes the presence of spatially correlated errors that are not captured by the independent variables, allowing for a more accurate representation of the data's spatial structure.

## 3. Results

### 3.1. Characteristics of the sample

Table 1 represents the percentage distribution of the sample size used in the present study. Approximately 52% of the sample consisted of female children, and the rest of the sample was male children. Among the sample, 13% of the sample

**Table 1. Sample distribution used in the analysis.**

| Variables | Sample | Percentage |
|---|---|---|
| **Preterm birth** | | |
| No | 2,01,657 | 87.45 |
| Yes | 28,933 | 12.55 |
| **Low birth weight** | | |
| No | 1,71,116 | 82.6 |
| Yes | 36,039 | 17.4 |
| **Sex of the child** | | |
| Male | 1,19,474 | 51.81 |
| Female | 1,11,116 | 48.19 |
| **Birth order** | | |
| 1 | 88,235 | 38.26 |
| 2 | 1,11,898 | 48.53 |
| 3+ | 30,457 | 13.21 |
| **Mother's age at delivery** | | |
| Below 20 | 26,282 | 11.4 |
| 20–24 | 98,137 | 42.56 |
| 25–29 | 68,494 | 29.7 |
| 30&above | 37,677 | 16.34 |
| **Place of delivery** | | |
| Home | 31,331 | 13.59 |
| Hospital | 1,99,259 | 86.41 |
| **Place of residence** | | |
| Rural | 1,84,038 | 79.81 |
| Urban | 46,552 | 20.19 |
| **Mother's body mass index** | | |
| Underweight | 42,112 | 18.73 |
| Normal | 1,42,504 | 63.38 |
| Overweight/obese | 40,208 | 17.88 |
| **Mother's education** | | |
| Illiterate/primary | 80,636 | 34.97 |
| Secondary | 1,18,602 | 51.43 |
| Higher | 31,352 | 13.6 |
| **Wealth status** | | |
| Poor | 1,16,744 | 50.63 |
| Middle | 44,700 | 19.39 |
| Rich | 69,146 | 29.99 |
| **Religion** | | |
| Hindu | 1,69,233 | 73.39 |
| Muslim | 33,194 | 14.4 |
| Others | 28,163 | 12.21 |
| **Cooking fuel** | | |
| Clean fuel | 97,582 | 42.32 |
| Solid fuel | 1,33,008 | 57.68 |

were born preterm, whereas 17% of the sample were LBW. About half of the sample belonged to 2nd birth order category. One out of 10 mothers were teenagers (under 20 years of age). About 14% of the children were born at home, and nearly 19% of the mothers were undernutrition or categorized as 'thin'. Half of the sample belonged to households classified as 'poor', and 73% of the sample belonged to the Hindu religion. Additionally, 58% of the households reported using solid fuels for cooking.

### 3.2. Spatial distribution of ABO and PM$_{2.5}$

The spatial distribution of ambient PM$_{2.5}$ shows a high concentration over the upper Gangetic region (Fig 1a), covering states like Uttar Pradesh, Bihar, Delhi, Punjab and Haryana, whereas a lower concentration was observed in the Southern and North-Eastern regions of India. The highest prevalence of PTB was observed in the Northern states, such as Himachal Pradesh (39%), Uttarakhand (27%) and Rajasthan (18%), and Delhi (17%), including North-Eastern states like Nagaland (S1 Table). In contrast, a lower prevalence of PTB was observed in Mizoram, Manipur and Tripura. As for LBW, the highest prevalence was in Punjab (22%), followed by Delhi, Dadra and Nagar Haveli, Madhya Pradesh, Haryana and Uttar Pradesh. Conversely, fewer children with LBW were observed in the North-East region of India (Fig 1b and Fig 1c).

### 3.3. Effects of PM$_{2.5}$ on ABO at the individual level

Fig 2 illustrates the weighted percentage of LBW and PTB by selected background characteristics. The prevalence of LBW was higher among females (20%) compared to males (17%), while PTB showed no significant gender difference (S2 Table). Teenage mothers exhibited the highest prevalence of both outcomes, with LBW rates decreasing with age but PTB rates rising after age 30. Children born at home or in rural areas had higher ABOs. Underweight mothers had the highest rates of LBW (22%) and PTB (13%), compared to 17% and 12%, respectively, among normal-weight mothers. Educational and socioeconomic disparities were evident, as illiterate or less-educated mothers and poorer households had greater ABOs. Mothers from Muslim households exhibited lower LBW rates but higher PTB prevalence. Additionally, households using solid fuels for cooking experienced higher rates of both LBW and PTB.

Our unadjusted and adjusted multivariate logistic regression models consistently revealed associations between air pollution during pregnancy and birth outcomes, as shown in Table 2. Higher ambient PM$_{2.5}$ concentrations during pregnancy were associated with higher odds of both LBW (AOR: 1.37; 95% CI: 1.29–1.45) and PTB (AOR: 1.67; 95% CI: 1.57–1.77). The odds ratios were higher in the unadjusted models for both LBW (OR: 1.56; 95% CI: 1.50–1.63) and PTB (OR: 1.62; 95% CI: 1.55–1.69). A slight increase in temperature was associated with higher odds of LBW (AOR: 1.03; 95% CI: 1.01–1.04), though it was not significantly associated with PTB, while higher rainfall was significantly associated with both LBW (AOR: 1.07; 95% CI: 1.03–1.12) and PTB (AOR: 1.04; 95% CI: 1.02–0.10). The use of solid fuel for cooking was associated with higher odds of LBW (AOR: 1.04; 95% CI: 1.01–1.07), but it was not significantly associated with PTB. Female children had higher odds of being born with LBW (AOR: 1.18; 95% CI: 1.15–1.20) compared to male children but had slightly lower odds of PTB (AOR: 0.97; 95% CI: 0.95–1.00). Higher birth order was associated with lower odds of LBW (AOR for second birth: 0.89; 95% CI: 0.86–0.91; AOR for third or higher birth: 0.84; 95% CI: 0.80–0.88). For PTB, only a third or higher birth order showed a significant decrease in odds (AOR: 0.94; 95% CI: 0.89–0.98). Teenage mothers (below 20 years) had higher odds of giving birth to children with LBW (AOR: 1.09; 95% CI: 1.04–1.15) and PTB (AOR: 1.08; 95% CI: 1.02–1.14). Children born at home had higher odds of LBW (AOR: 1.11; 95% CI: 1.06–1.16). Urban residence was associated with higher odds of LBW (AOR: 1.07; 95% CI: 1.04–1.11) and lower odds of PTB (AOR: 0.96; 95% CI: 0.92–0.99). Underweight mothers had higher odds of both LBW (AOR: 1.27; 95% CI: 1.24–1.31) and PTB (AOR: 1.04; 95% CI: 1.01–1.08), while overweight or obese mothers had lower odds of both LBW (AOR: 0.93; 95% CI: 0.90–0.96) and PTB (AOR: 0.95; 95% CI: 0.92–0.99). Lower maternal education levels were associated with higher odds of LBW (AOR for illiterate/primary: 1.40; 95% CI: 1.34–1.46; AOR for secondary: 1.23; 95% CI: 1.18–1.28), while higher education was associated with lower odds of PTB (AOR: 0.95; 95% CI: 0.91–0.99). Poor mothers had higher odds of LBW (AOR: 1.09;

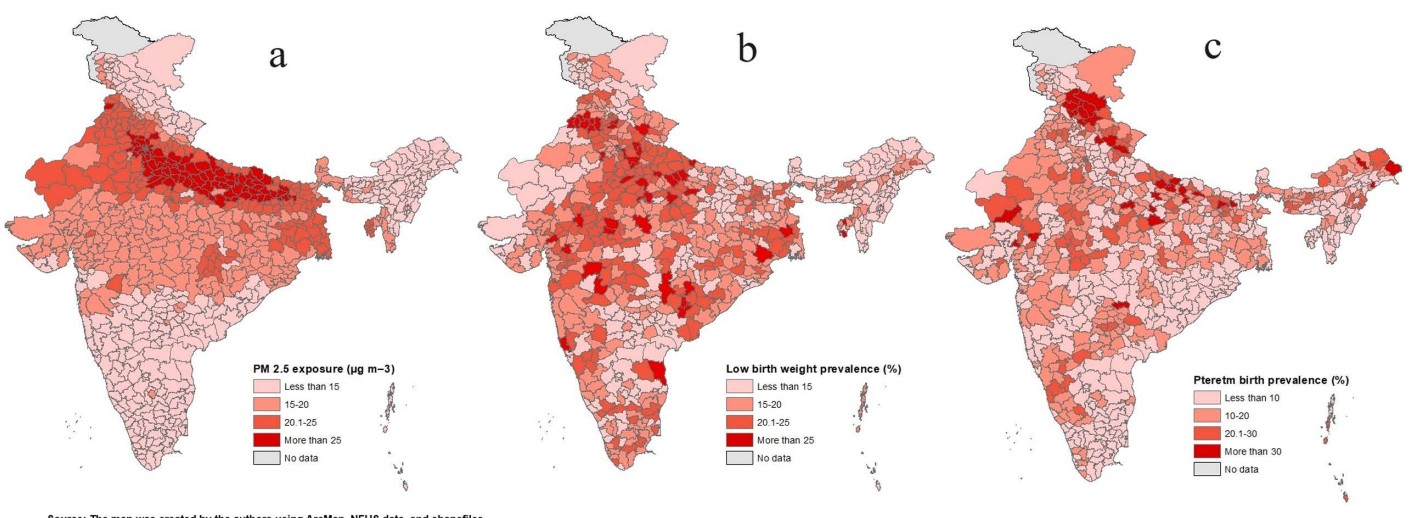

**Fig 1. (a) Spatial distribution of in-utero exposure to PM$_{2.5}$, (b) low birth weight and (c) preterm birth across in India.** Source: The map was created by the authors using ArcMap, NFHS data, and shapefiles.

95% CI: 1.05–1.13) and lower odds of PTB (AOR: 0.92; 95% CI: 0.88–0.95). Muslim mothers had lower odds of LBW (AOR: 0.87; 95% CI: 0.84–0.91) and similar odds of PTB compared to Hindu mothers.

There was a high likelihood of LBW in women who experienced high levels of PM$_{2.5}$ during their gestational period (Fig 3). Employing a distributed spline approach, the study identified a growing trend in the risk of delivering LBW babies as the exposure level increased, especially after the exposure level of 40 PM$_{2.5}$ ug/m$^3$. Concerning PTB, the odds displayed a rapid and exponential increase in relation to the PM$_{2.5}$ exposures and the increasing trend became notably rapid after reaching the exposure level of 50 PM$_{2.5}$ ug/m$^3$.

### 3.4. Sensitivity analysis

The unadjusted analysis (Table 3) demonstrated a significantly increased risk of LBW among women exposed to higher levels of PM$_{2.5}$ during pregnancy (OR: 1.56, 95% CI: 1.50-1.63). However, after adjusting for a comprehensive set of confounders including environmental, socioeconomic, maternal, and child factors, the strength of this association attenuated. Conversely, for PTB, the adjusted odds ratio increased slightly compared to the unadjusted model. These contrasting findings emphasize the crucial role of rigorous adjustment for potential confounders in accurately assessing the impact of PM$_{2.5}$ exposure on these ABOs. These sensitivity analyses further strengthen the confidence in our findings, suggesting that the observed associations between PM$_{2.5}$ exposure and ABOs are robust.

### 3.5. Spatial association between PM$_{2.5}$ and ABO

The findings of the bivariate LISA map showed that children living in the Northern districts of India faced a higher vulnerability to ambient air pollution as the high-high clusters of spatial association were found in Punjab, Delhi, Madhya Pradesh, Rajasthan and some parts of Uttar Pradesh (Fig 4). A total of 109 districts had a significant association between in-utero exposure to PM$_{2.5}$ and LBW. On the other hand, 40 districts had high-high clusters of autocorrelations between PM$_{2.5}$ and PTB. Notably, it was observed that most of the districts of Uttar Pradesh were found to be more vulnerable to PM$_{2.5}$ in the context of preterm birth.

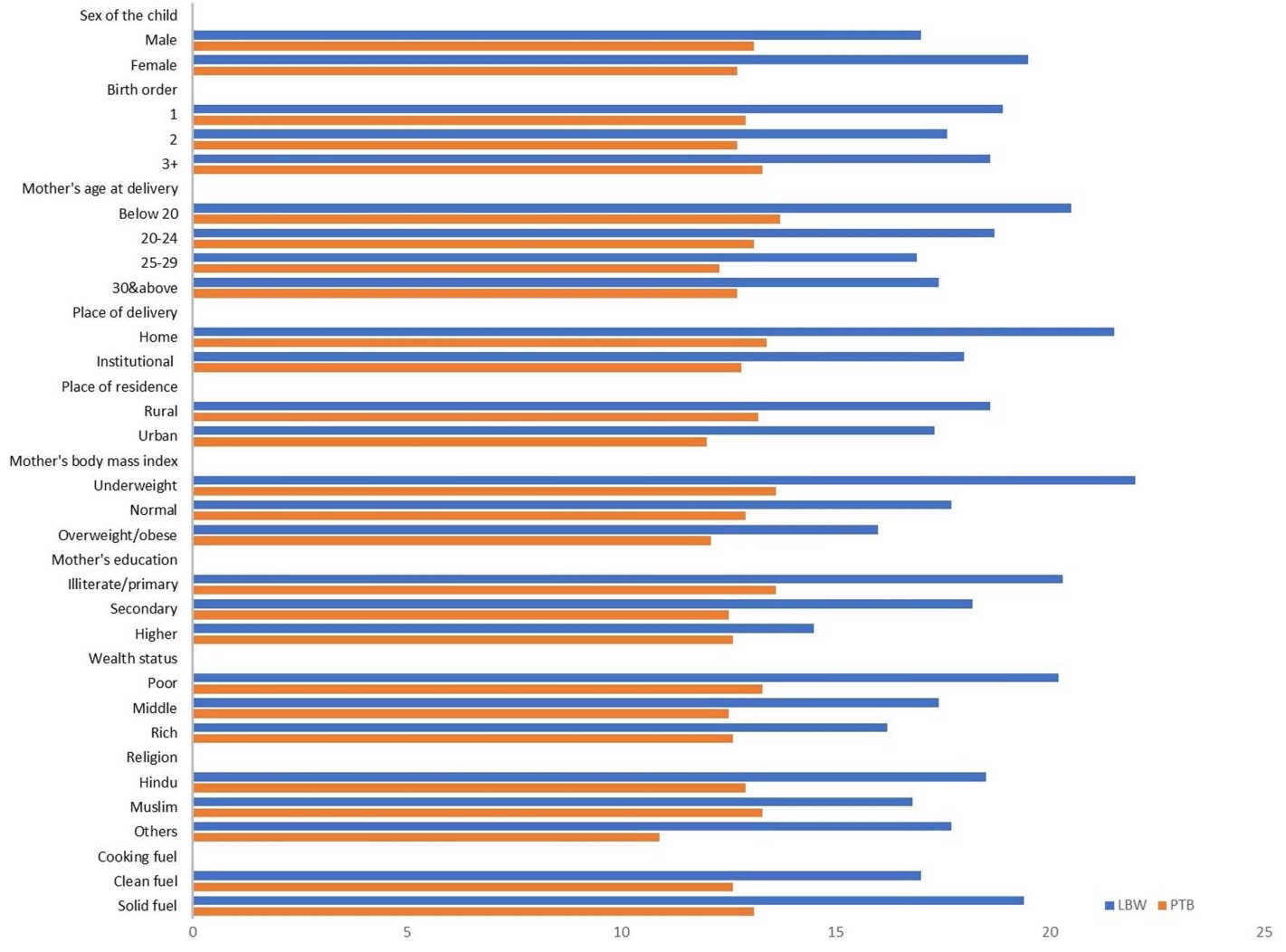

**Fig 2. Weighted percentage of low birth weight and preterm birth by background characteristics.**

Additionally, the study employed OLS, GWR, MGWR, SLM, and SEM, to investigate the spatial relationship between $PM_{2.5}$ exposures during pregnancy and the occurrences of LBW and PTB, while considering potential confounding factors. In line with our hypothesis, the analysis revealed a robust spatial association between $PM_{2.5}$ exposures and LBW, although the strength of this association varied across the different models. Notably, the GWR model showed the highest impact of $PM_{2.5}$ exposures ($\beta = 0.122$, SE = 0.009), while MGWR demonstrated the highest goodness-of-fit ($R^2 = 68\%$) (Table 4). Regarding PTB, the MGWR model provided the best explanatory power, accounting for 54% of the variance, surpassing other models. In the GWR model, a noteworthy finding was that a 10 μg m−3 increase in $PM_{2.5}$ exposure corresponded to a 5% increase in the prevalence of LBW and a 12% increase in PTB, with similar values observed in the other models (Table 5). Overall, a reduction in the prevalence of LBW and PTB was observed across all models, with MGWR and GWR models showing particularly promising outcomes. Nevertheless, higher prevalence rates persisted in Northern India and certain parts of the eastern region, most notably in Odisha (Fig 5).

**Table 2. Adjusted and unadjusted odds ratio showing the effects of in-utero exposure to PM$_{2.5}$ on low birth weight and preterm birth.**

| Determinants | Low birth weight | | Preterm birth | |
|---|---|---|---|---|
| | OR | AOR | OR | AOR |
| **PM$_{2.5}$ (µg/m³)** | | | | |
| <mean | | | | |
| >= mean | 1.56***(1.50 1.63) | 1.37***(1.29 1.45) | 1.62***(1.55 1.69) | 1.67***(1.57 1.77) |
| **Temperature (°C)** | | | | |
| <mean | | | | |
| >= mean | | 1.14***(1.01 1.04) | | 0.98 (0.96 1.01) |
| **Rainfall (mm)** | | | | |
| <mean | | | | |
| >= mean | | 1.07***(1.03 1.12) | | 1.04***(1.02 1.10) |
| **Sex of the child** | | | | |
| Male (Ref) | | | | |
| Female | | 1.18***(1.15 1.20) | | 0.97**(0.95 1) |
| **Birth order** | | | | |
| 1 (Ref) | | | | |
| 2 | | 0.89***(0.86 0.91) | | 0.99 (0.96 1.02) |
| 3+ | | 0.84***(0.8 0.88) | | 0.94***(0.89 0.98) |
| **Mother's age at delivery** | | | | |
| Below 20 | | 1.09***(1.04 1.15) | | 1.08***(1.02 1.14) |
| 20-24 | | 1.03 (0.99 1.07) | | 1.03 (0.98 1.07) |
| 25-29 | | 1 (0.96 1.03) | | 0.97 (0.93 1.01) |
| 30&above (Ref) | | | | |
| **Place of delivery** | | | | |
| Home (Ref) | | 1.11***(1.06 1.16) | | 1.00 (0.97 1.04) |
| Hospital | | | | |
| **Place of residence** | | | | |
| Rural (Ref) | | | | |
| Urban | | 1.07***(1.04 1.11) | | 0.96**(0.92 0.99) |
| **Mother's body mass index** | | | | |
| Normal (Ref) | | | | |
| Underweight | | 1.27***(1.24 1.31) | | 1.04***(1.01 1.08) |
| Overweight/obese | | 0.93***(0.9 0.96) | | 0.95***(0.92 0.99) |
| **Mother's education** | | | | |
| Illiterate/primary | | 1.40***(1.34 1.46) | | |
| Secondary | | 1.23***(1.18 1.28) | | 1.00 (0.95 1.04) |
| Higher (Ref) | | | | 0.95**(0.91 0.99) |
| **Wealth status** | | | | |
| Poor | | 1.09***(1.05 1.13) | | 0.92***(0.88 0.95) |
| Middle | | 0.98 (0.94 1.02) | | 0.95**(0.92 0.99) |
| Rich (Ref) | | | | |
| **Religion** | | | | |
| Hindu (Ref) | | | | |
| Muslim | | 0.87***(0.84 0.91) | | 0.97 (0.94 1.01) |
| Others | | 0.70***(0.67 0.73) | | 0.88***(0.84 0.92) |

*(Continued)*

**Table 2.** (Continued)

| Determinants | Low birth weight | | Preterm birth | |
| --- | --- | --- | --- | --- |
| | OR | AOR | OR | AOR |
| **Cooking fuel** | | | | |
| Clean fuel (Ref) | | | | |
| Solid fuel | | 1.04 ***(1.01 1.07) | | 0.94 (0.91 1.03) |

Ref: Reference category; *** p<0.001, **<0.05, * p<0.1.

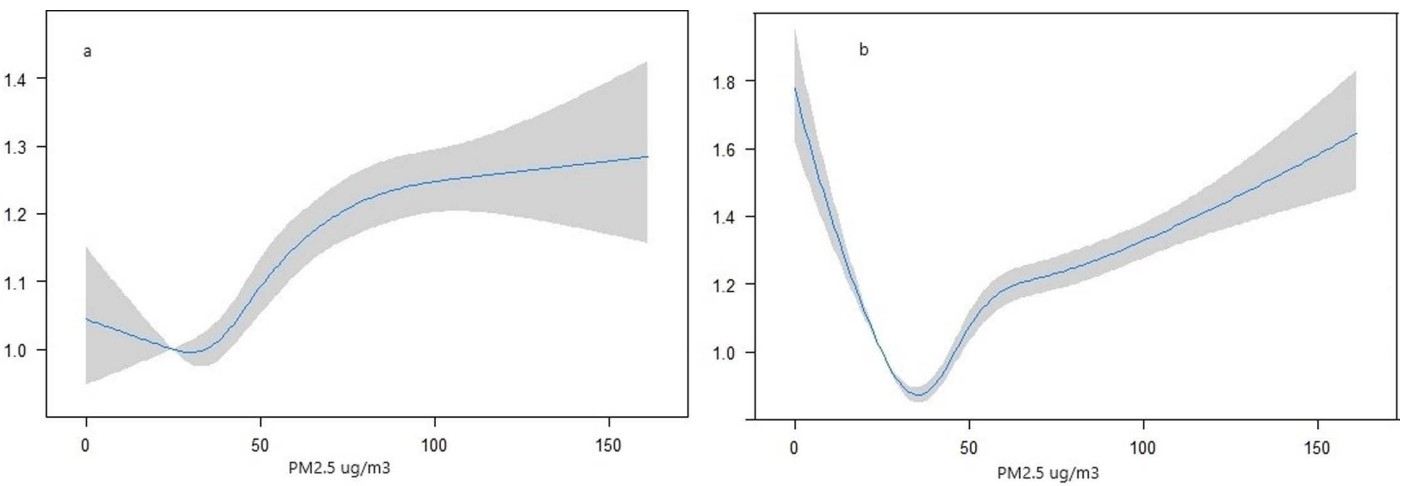

**Fig 3. Susceptibility to (a) low birth weight, and (b) preterm birth due to the in-utero exposure to PM$_{2.5}$.**

## 4. Discussion

The study is the first attempt to measure PTB at the Indian district level by exploring the calendar data of NFHS [48]. The analysis establishes the evidence of the association between in-utero exposure to PM$_{2.5}$ and ABOs by leveraging satellite data and large-scale survey data. The individual-level analysis reveals that an increase in ambient PM$_{2.5}$ is associated with a greater likelihood of LBW and PTB, consistent with previous studies [25,53–55]. We observe a higher value of the odds ratio of having preterm infants whose mothers were exposed to PM$_{2.5}$ during pregnancy as compared to the results for LBW. A significant role of indoor air pollution has been observed in the study for LBW. Further, climatic factors such as rainfall and temperature are significantly associated with ABOs in India. To the best of our knowledge, this study is the first of its kind to understand the association between in-utero exposure to PM$_{2.5}$ and ABOs in India.

It is worth noting that except for Nagaland, North-Eastern states had a lower prevalence of PTB and low birth weight, coinciding with lower pollution levels in that region. Previous studies have found that the concentration of ambient PM$_{2.5}$ is highest in the states of upper-Gangetic plains such as Delhi, Punjab, Haryana, Uttar Pradesh and Bihar, as observed in the present study [56–58]. A recent Lancet study suggests that the average PM$_{2.5}$ concentrations in Delhi were 13.8 times higher than that in Kerala [13]. The Northern states had the highest PM$_{2.5}$ concentrations. The NFHS-5 report suggests that a large proportion of households in the Northern parts of India use solid fuels compared to other regions [39]. It is well documented that the residential sector is a significant contributor to the total PM$_{2.5}$ emissions along with the industry,

Table 3. Sensitivity analysis of the effects of PM$_{2.5}$ on low birth weight and preterm birth.

| Adjusted characteristic | Low birth weight | | | | Preterm birth | | | |
|---|---|---|---|---|---|---|---|---|
| | Odds ratio | P value | 95% CI | | Odds ratio | P value | 95% CI | |
| | | | Lower | Upper | | | Lower | Upper |
| **Unadjusted** | 1.56 | <0.00 | 1.50 | 1.63 | 1.62 | <0.00 | 1.55 | 1.69 |
| Temperature (°C) | 1.46 | <0.00 | 1.40 | 1.52 | 1.69 | <0.00 | 1.62 | 1.76 |
| Climatic factors (temperature + rainfall) | 1.39 | <0.00 | 1.33 | 1.45 | 1.57 | <0.00 | 1.50 | 1.65 |
| Cooking fuel | 1.53 | <0.00 | 1.47 | 1.60 | 1.63 | <0.00 | 1.57 | 1.71 |
| Child characteristics | 1.59 | <0.00 | 1.51 | 1.68 | 1.76 | <0.00 | 1.67 | 1.87 |
| Mothers' characteristics | 1.52 | <0.00 | 1.45 | 1.58 | 1.61 | <0.00 | 1.54 | 1.69 |
| Socioeconomic factors | 1.48 | <0.00 | 1.42 | 1.54 | 1.59 | <0.00 | 1.52 | 1.66 |
| Socioeconomic, child and mothers' Characteristics | 1.45 | <0.00 | 1.37 | 1.54 | 1.70 | <0.00 | 1.60 | 1.80 |
| All the determinants | 1.37 | <0.00 | 1.29 | 1.45 | 1.67 | <0.00 | 1.57 | 1.77 |

energy and agriculture sectors [59]. Among industrial, residential and energy sources, the contribution of energy sources to total emissions is the maximum, while residential sources contribute the maximum to PM$_{2.5}$ emissions during winter and post-monsoon [59]. In contrast, certain studies conducted on future emissions scenarios in India forecast an increase in PM$_{2.5}$ levels [60]. However, at the urban or city level, where most households are already using cleaner fuel, reducing vehicular emissions (both exhaust and non-exhaust) emerges as a crucial strategy for reducing PM$_{2.5}$ levels. It was prominently observed during the coronavirus pandemic (COVID-19) lockdown in Indian cities when traffic reduction substantially minimized urban areas' exposure to air pollutants [56]. Moreover, the issue of air pollution is exacerbated by crop residue burning and forest fires in northern India, significantly contributing to the toxic air quality in the region [61,62]. The higher prevalence of LBW and PTB in the districts of the Northern region indicates a spatial association between PM$_{2.5}$ and birth outcomes. In line with our hypothesis, the study finds an individual level and spatial association between in-utero exposure to PM$_{2.5}$ and LBW and PTB.

Medical studies found that the mother's fetus grows rapidly in the third trimester of pregnancy [63]. That period is more sensitive, and exposure to PM$_{2.5}$ can hemorrhage the foetal growth easily [64]. Nevertheless, the level of thyroid hormone might be affected by the toxicity of PM$_{2.5}$, which is a responsible factor for less fetus weight [65]. Previous studies have found that fetal growth depends on many factors, such as mothers' health, socioeconomic condition and genetic factors [66,67]. Thus, the present study has adjusted for those potential plausible factors. Nevertheless, the study establishes a significant association between PM$_{2.5}$ and LBW. Further investigation is required to explore the biological mechanism by considering the biomarker measurement of the fetus. Sometimes inhaled PM$_{2.5}$ penetrates the toxic gases that damage deoxyribonucleic acid, restricting nutrient supply to the fetus resulting in pregnancy complications during the last trimester of pregnancy that triggers the possibility of having a PTB [68,69]. Moreover, substantial exposure to a large amount of PM$_{2.5}$ has been linked to fetal malformation, miscarriage and stillbirth, all of which can influence subsequent birth outcomes [70–72]. Understanding the impact of exposure to indoor air pollution during pregnancy is also important because pregnant women tend to spend most of their time indoors and this time only increases once the pregnancy progresses [73]. Past studies found that releasing pollutants from uncleaned biomass burning in the household restricts fetal growth, increasing the probability of having a child with low weight and PTB [74,75]. Since 1970, the Indian government has been trying to improve child health by adopting several schemes that promote the utilization of maternal healthcare facilities, spread awareness of reproductive health, supply nutrients to mothers and newborns, etc. ABOs not only impact a child's health but also reduce the productivity of the human resources of a country.

Climatic factors, particularly temperature and precipitation, have significant associations with ABOs in India through multiple biological and environmental pathways. Elevated temperatures have been linked to maternal heat stress,

**Fig 4. Bivariate LISA map showing the spatial association between in-utero exposure to PM$_{2.5}$ and preterm birth (a) and low birth weight (b).** Source: The map was created by the authors using ArcMap, NFHS data, and shapefiles.

dehydration, and increased cardiovascular strain, which can impair uteroplacental blood flow, disrupt fetal development, and elevate the risk of LBW and PTB [76,77]. Heat-induced oxidative stress and systemic inflammation may further compromise placental function, leading to pregnancy complications [78]. Prolonged exposure to extreme heat can also alter maternal endocrine function, increasing the risk of gestational hypertension and preeclampsia, which are known to contribute to fetal growth restriction and premature delivery [76]. Furthermore, high ambient temperatures can exacerbate maternal dehydration, affecting amniotic fluid levels and increasing the likelihood of PTB [79,80]. Excessive rainfall, particularly during the monsoon season, heightens exposure to waterborne and vector-borne pathogens, increasing

**Table 4. Spatial regression models showing the spatial association between PM$_{2.5}$ and preterm birth adjusted confounding factors.**

| Variables | OLS | | GWR | | MGWR | | SLM | | SEM | |
|---|---|---|---|---|---|---|---|---|---|---|
| | Coefficient | SE | Coefficient | SE | Coefficient | SE | Coefficient | SE | Coefficient | SE |
| **PM$_{2.5}$ (µg/m³)** | 0.059 | 0.009 | 0.122 | 0.009 | 0.119 | 0.016 | 0.043 | 0.014 | 0.031 | 0.008 |
| Underweighted mother | 0.159 | 0.031 | 0.179 | 0.007 | 0.163 | 0.003 | 0.124 | 0.032 | 0.102 | 0.027 |
| Illiterate mother | 0.061 | 0.018 | 0.095 | 0.009 | 0.013 | 0.001 | 0.068 | 0.021 | 0.040 | 0.016 |
| Belong to poor family | -0.053 | 0.015 | 0.234 | 0.008 | 0.494 | 0.000 | -0.011 | 0.019 | -0.022 | 0.013 |
| Hindu religion | 0.010 | 0.009 | -0.055 | 0.006 | -0.017 | 0.002 | 0.007 | 0.011 | -0.002 | 0.008 |
| Non-institutional delivery | -0.083 | 0.025 | -0.067 | 0.008 | -0.155 | 0.000 | -0.060 | 0.027 | -0.057 | 0.022 |
| Using solid cooking fuel | 0.062 | 0.014 | 0.126 | 0.009 | -0.118 | 0.000 | 0.024 | 0.017 | 0.028 | 0.013 |
| Belong to rural areas | -0.018 | 0.012 | -0.137 | 0.008 | -0.114 | 0.005 | 0.002 | 0.012 | -0.001 | 0.011 |
| Female child | -0.005 | 0.058 | 0.003 | 0.003 | 0.001 | 0.004 | 0.008 | 0.048 | 0.001 | 0.050 |
| Teen aged mothers | 0.022 | 0.034 | 0.066 | 0.007 | 0.101 | 0.007 | 0.023 | 0.040 | 0.005 | 0.030 |
| Having 3& more births | -0.090 | 0.046 | -0.122 | 0.007 | -0.03 | 0.005 | -0.102 | 0.048 | -0.069 | 0.040 |
| Average rainfall (°C) | 0.000 | 0.003 | -0.071 | 0.005 | -0.055 | 0.000 | -0.003 | 0.003 | 0.001 | 0.003 |
| Average temperature (mm) | 0.041 | 0.041 | 0.026 | 0.007 | 0.165 | 0.000 | 0.105 | 0.052 | 0.023 | 0.036 |
| R² | 0.34 | | | | 0.68 | | 0.49 | | 0.52 | |
| **AIC** | 4192.36 | | | | 1420.83 | | 4045.78 | | 4022.85 | |

**Table 5. Spatial regression models showing the spatial association between PM$_{2.5}$ and low birth weight adjusted confounding factors.**

| Variables | OLS | | GWR | | MGWR | | SLM | | SEM | |
|---|---|---|---|---|---|---|---|---|---|---|
| | Coefficient | SE | Coefficient | SE | Coefficient | SE | Coefficient | SE | Coefficient | SE |
| **PM$_{2.5}$ (µg/m³)** | 0.043 | 0.016 | 0.053 | 0.012 | 0.041 | 0.018 | 0.018 | 0.014 | 0.035 | 0.024 |
| Underweighted mother | 0.009 | 0.053 | 0.051 | 0.014 | 0.044 | 0.012 | -0.022 | 0.046 | -0.059 | 0.056 |
| Illiterate mother | 0.035 | 0.030 | 0.169 | 0.011 | 0.214 | 0.000 | 0.030 | 0.026 | 0.070 | 0.036 |
| Belong to poor family | -0.055 | 0.026 | -0.121 | 0.018 | -0.140 | 0.000 | -0.024 | 0.022 | -0.029 | 0.033 |
| Hindu religion | 0.050 | 0.016 | 0.078 | 0.010 | 0.038 | 0.003 | 0.023 | 0.014 | 0.023 | 0.019 |
| Non-institutional delivery | 0.050 | 0.043 | 0.047 | 0.019 | 0.071 | 0.006 | 0.022 | 0.037 | 0.008 | 0.046 |
| Using solid cooking fuel | -0.014 | 0.025 | -0.096 | 0.015 | -0.045 | 0.005 | -0.019 | 0.021 | -0.048 | 0.030 |
| Belong to rural areas | 0.044 | 0.021 | 0.054 | 0.005 | 0.062 | 0.000 | 0.027 | 0.018 | 0.042 | 0.021 |
| Female child | 0.032 | 0.100 | 0.031 | 0.009 | 0.045 | 0.004 | 0.030 | 0.086 | 0.038 | 0.084 |
| Teen aged mothers | 0.028 | 0.059 | 0.022 | 0.005 | 0.022 | 0.001 | 0.019 | 0.051 | 0.027 | 0.069 |
| Having 3& more births | 0.053 | 0.078 | 0.096 | 0.009 | 0.05 | 0.013 | 0.014 | 0.068 | 0.003 | 0.082 |
| Average rainfall (°C) | 0.000 | 0.005 | 0.030 | 0.008 | 0.030 | 0.003 | 0.000 | 0.004 | -0.001 | 0.005 |
| Average temperature (mm) | -0.291 | 0.070 | -0.143 | 0.013 | -0.112 | 0.009 | -0.117 | 0.061 | -0.106 | 0.089 |
| R² | 0.17 | | | | 0.54 | | 0.29 | | 0.31 | |
| **AIC** | 4953.57 | | | | 1645.21 | | 4810.00 | | 4796.55 | |

maternal infection risk, which has been associated with intrauterine growth restriction and PTB [81]. Heavy rainfall and flooding can also cause displacement, disrupt healthcare access, and reduce antenatal care utilization, leading to delayed or inadequate medical attention during pregnancy [82]. In flood-prone regions, pregnant women face increased risks of malnutrition due to food shortages, as well as heightened psychological stress, both of which are linked to poor fetal growth and adverse perinatal outcomes [83]. Additionally, extreme precipitation events can lead to increased contamination of drinking water sources, exacerbating the risk of diarrheal diseases and other infections that can negatively impact maternal and fetal health [84]. These findings highlight the urgent need for climate-resilient maternal healthcare policies

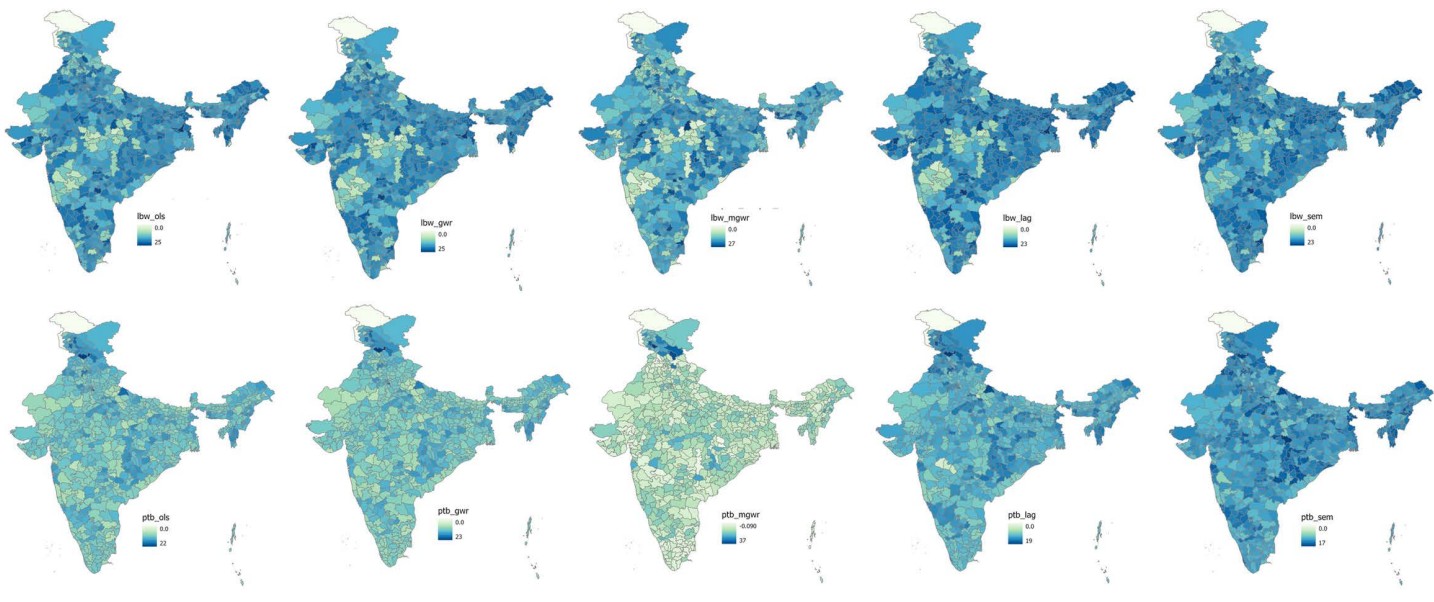

**Fig 5. Predicted low birth weight (LBW) and preterm birth (PTB) results from Ordinary List Square (OLS), Geographically Weighed Regression (GWR), Multiscale Geographically Weighed Regression (MGWR), Spatial Lag Model (SLM) and Spatial Error Model (SEM) after adjusting environmental, Socioeconomic maternal and child characteristics.** Source: The map was created by the authors using ArcMap, NFHS data, and shapefiles.

and interventions. Strategies such as heat adaptation measures for pregnant women, improved early warning systems for extreme weather events, and enhanced healthcare accessibility during monsoon seasons are essential to mitigating the impact of temperature and rainfall extremes on maternal and neonatal health in India.

### 4.1. Limitations

This study assumes that mothers did not change their residence from pregnancy until the survey date, which may introduce spatial uncertainty. Some of the confounding factors has not been included in the analysis due to underreporting and not available in the dataset. Additionally, reliance on self-reported birth weight and pregnancy duration introduces potential recall and reporting biases. Given the cross-sectional nature of the study, causal relationships cannot be established. Future research using longitudinal designs could explore seasonality effects and provide more robust insights. However, a deeper understanding of the biological mechanisms underlying the relationship between air pollution and malnutrition requires further epidemiological research, which is beyond the scope of existing datasets.

### 5. Conclusion

The present study provides robust evidence linking in-utero exposure to ambient $PM_{2.5}$ and climatic factors, such as temperature and rainfall, with ABOs in India. The geostatistical analysis underscores the need for targeted interventions, particularly in Northern districts identified as highly vulnerable. To address these challenges, comprehensive strategies are essential. The geostatistical models can be utilized in future studies to support area-specific policy development by identifying zones that are more vulnerable. The National Clean Air Program should be intensified, with stricter emission standards and enhanced air quality monitoring. Integrating air quality data with health surveillance systems will enable precise

identification of at-risk populations. Additionally, policies promoting clean cooking fuels and energy-efficient technologies can reduce indoor air pollution. Climate adaptation strategies, such as developing heat action plans and improving water management, should be incorporated into public health planning to mitigate the effects of extreme temperatures and irregular rainfall. Public health initiatives must raise awareness about the risks of air pollution and climate change, particularly among pregnant women.

## Supporting information

**S1 Table. The weighted prevalence of low birth weight and preterm birth across Indian states and union territories, 2019–21.**
(DOCX)

**S2 Table. Weighted percentage of low birth weight and preterm birth by background characteristics.**
(DOCX)

## Author contributions

**Conceptualization:** Arup Jana, Malay Pramanik, Arabinda Maiti, Aparajita Chattopadhyay, Mary Abed Al Ahad.

**Data curation:** Arup Jana.

**Formal analysis:** Arup Jana.

**Investigation:** Arup Jana, Malay Pramanik, Arabinda Maiti, Aparajita Chattopadhyay.

**Methodology:** Arup Jana, Malay Pramanik, Arabinda Maiti, Aparajita Chattopadhyay.

**Project administration:** Arup Jana.

**Resources:** Arup Jana, Mary Abed Al Ahad.

**Software:** Arup Jana.

**Validation:** Arup Jana.

**Visualization:** Arup Jana.

**Writing – original draft:** Arup Jana, Malay Pramanik, Arabinda Maiti, Aparajita Chattopadhyay.

**Writing – review & editing:** Arup Jana, Malay Pramanik, Arabinda Maiti, Aparajita Chattopadhyay, Mary Abed Al Ahad.

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
