## [Decision Letter · Decision Letter 0]

PGPH-D-24-02116

In-utero exposure to PM2.5 and adverse birth outcomes in India: Geostatistical modelling using remote sensing and demographic health survey data 2019-21

Dear Dr. Abed Al Ahad,

Thank you for submitting your manuscript to PLOS Global Public Health. After careful consideration, we feel that it has merit but does not fully meet PLOS Global Public Health’s publication criteria as it currently stands. Therefore, we invite you to submit a revised version of the manuscript that addresses the points raised during the review process.

We look forward to receiving your revised manuscript.

Kind regards,

Bhargav Krishna

Academic Editor

Journal Requirements:

1. We noticed you have some minor occurrence of overlapping text with the following previous publication(s), which needs to be addressed:

https://www.nature.com/articles/s41370-023-00591-5’

https://www.researchgate.net/publication/341944202_Associations_between_green_space_and_preterm_birth_Windows_of_susceptibility_and_interaction_with_air_pollution

In your revision ensure you cite all your sources (including your own works), and quote or rephrase any duplicated text outside the methods section. Further consideration is dependent on these concerns being addressed.

2. Figures 1, 4 and 5: please (a) provide a direct link to the base layer of the map (i.e., the country or region border shape) and ensure this is also included in the figure legend; and (b) provide a link to the terms of use / license information for the base layer image or shapefile. We cannot publish proprietary or copyrighted maps (e.g. Google Maps, Mapquest) and the terms of use for your map base layer must be compatible with our CC-BY 4.0 license. 

Additional Editor Comments (if provided):

Reviewers' comments:

Reviewer's Responses to Questions

**Comments to the Author**

1. Does this manuscript meet PLOS Global Public Health’s publication criteria ? Is the manuscript technically sound, and do the data support the conclusions? The manuscript must describe methodologically and ethically rigorous research with conclusions that are appropriately drawn based on the data presented.

Reviewer #1: Yes

Reviewer #2: Yes

2. Has the statistical analysis been performed appropriately and rigorously?

Reviewer #1: Yes

Reviewer #2: Yes

3. Have the authors made all data underlying the findings in their manuscript fully available (please refer to the Data Availability Statement at the start of the manuscript PDF file)?

Reviewer #1: Yes

Reviewer #2: Yes

4. Is the manuscript presented in an intelligible fashion and written in standard English?

Reviewer #1: Yes

Reviewer #2: Yes

5. Review Comments to the Author

Reviewer #1: Please refer to my detailed comments in the PDF version of the paper. My recommendation is to publish this article after revision. The findings and discussion section need significant improvement - improving the presentation of results in the findings section which is currently very "listy" and hard to follow for the reader, and the discussion section needs to significantly expand on the mechanisms and context driving the results.

Reviewer #2: General comments:

The manuscript presents a significant and timely topic on examining the relationship between PM2.5 air pollution and adverse birth outcomes (ABOs) in India. The study employs robust data from the National Family Health Survey (NFHS) combined with satellite-based PM2.5 and climatic data, offering a unique geostatistical approach to identify vulnerable populations at individual and district levels. The findings—highlighting associations between PM2.5 exposure and higher risks of low birth weight (LBW) and preterm birth (PTB)—are compelling and policy-relevant. However, the manuscript has some methodological and interpretive limitations that must be acknowledged in the Discussion before it can be considered for publication.

Major comments:

1. The methods section does not specify whether non-linear relationships between PM2.5 exposure and birth outcomes were considered. If only linear models were used, the analysis may fail to capture critical thresholds at very high or low exposure levels. This should be included in the discussion as a potential limitation.

2. The reliance on self-reported birth outcomes in NFHS may introduce recall bias. This should be acknowledged as another potential limitation.

3. The study uses annual average PM2.5 concentrations derived from satellite data. This approach does not account for within-year variations in exposure, such as seasonal fluctuations, which could influence pregnancy outcomes. The authors should discuss how this temporal resolution might misclassify exposure, particularly in areas with distinct pollution patterns across seasons.

4. While temperature and rainfall are included, the methods do not clarify whether these variables were spatially or temporally aligned with PM2.5 exposure or birth outcome data. Mismatched temporal or spatial aggregation could introduce exposure misclassification and bias the findings. Additional details should be provided and discussed in detail.

5. Although the authors mention smoking and alcohol use as unmeasured confounders, they do not acknowledge other potential behavioral confounders, such as maternal physical activity, diet, and access to prenatal care, which may vary spatially and interact with air pollution exposure. Please include these.

Minor comments:

6. Across the manuscript, please mention that the observed effects are per unit increase in PM2.5.

7. In the abstract, please mention the abbreviations in full as they first appear in text: LBW, AOR, PM2.5, MGWR.

8. Effects of rainfall and temperature are only briefly mentioned in abstract. These should be expanded to provide the full context of these findings.

9. Correct the “mu” symbol in the unit of measurement of PM2.5 throughout the manuscript.

10. Lines 98-99 in the Introduction suggest that only LBW and PTB have been associated with air pollution in the existing literature. There are other adverse birth outcomes such as spontaneous abortion for which epidemiological evidence exists. Please correct the sentence to be more inclusive.

6. PLOS authors have the option to publish the peer review history of their article (what does this mean? ). If published, this will include your full peer review and any attached files.

**Do you want your identity to be public for this peer review?** For information about this choice, including consent withdrawal, please see our Privacy Policy .

Reviewer #1: No

Reviewer #2: **Yes: ** Rachit Sharma

---

## [Editor Report · Decision Letter 1]

In-utero exposure to PM2.5 and adverse birth outcomes in India: Geostatistical modelling using remote sensing and demographic health survey data 2019-21

PGPH-D-24-02116R1

Dear Dr Abed Al Ahad,

We are pleased to inform you that your manuscript 'In-utero exposure to PM2.5 and adverse birth outcomes in India: Geostatistical modelling using remote sensing and demographic health survey data 2019-21' has been provisionally accepted for publication in PLOS Global Public Health.

Best regards,

Bhargav Krishna

Academic Editor